# Inhibitory Effects against Alpha-Amylase of an Enriched Polyphenol Extract from Pericarp of Mangosteen (*Garcinia mangostana*)

**DOI:** 10.3390/foods11071001

**Published:** 2022-03-29

**Authors:** Xiaofang Li, Haoze Chen, Yan Jia, Jinming Peng, Chunmei Li

**Affiliations:** 1College of Food Science and Technology, Huazhong Agricultural University, Wuhan 430070, China; xiaofang_li@webmail.hzau.edu.cn (X.L.); chenhaoze1998@163.com (H.C.); 2Beijing Key Lab of Plant Resource Research and Development, School of Science, Beijing Technology and Business University, Beijing 100048, China; 3College of Food Science and Technology, Zhongkai University of Agriculture and Engineering, Guangzhou 510225, China; pengjmiyz@163.com; 4Key Laboratory of Environment Correlative Food Science, Ministry of Education, Huazhong Agricultural University, Wuhan 430070, China

**Keywords:** Mangosteen (*Garcinia mangostana*), polyphenol, α-amylase inhibition, fluorescence quenching, postprandial blood glucose level

## Abstract

The pericarp of mangosteen, a by-product of the mangosteen, is rich in polyphenols. In this study, an efficient and environmentally friendly method for preparative enrichment of polyphenols from mangosteen pericarp (MPPs) was developed, and the inhibitory effects on starch digestion were also evaluated. It was found that the optimal extract method of MPPs was at a solid to solvent ratio of 1:50 g/mL, pH of 2, and at 80 °C for 2 h. The IC_50_ of MPPs for α-amylase was 0.28 mg/mL. Based on the fluorescence quenching results, we presumed that MPPs could alter the natural structure of α-amylase, resulting in inhibitory activity on α-amylase. In addition, MPPs significantly reduced the blood glucose peak and AUC of glucose responses in rats after ingestion of the starch solution. Taken together, MPPs may have the potential as a functional supplement for blood glucose control and diabetes prevention.

## 1. Introduction

Diabetes is a chronic metabolic disease characterized by hyperglycemia and the most common is type 2 diabetes mellitus (T2DM). In 2019, about 463 million adults worldwide lived with diabetes (1 of 11 people were diabetic) and about 4.2 million deaths were attributed to diabetes or the relative complications [1]. According to the IDF Diabetes Atlas (2019), the three countries with the highest number of adults with diabetes are China, India, and the United States and this is expected to remain so to 2030. Many factors influence the development of T2DM. Healthy lifestyle behaviors have been proven to be the most cost-effective precautionary measures for inhibition of the development of T2DM [2,3]. Alpha-amylase is the key enzyme responsible for digestion of starch and is also one of the key molecular targets for the treatment of T2DM [4]. The potential inhibitors of α-amylase may slow down carbohydrates digestion and reduce the glucose absorption rate, thus, lowering the postprandial blood glucose level, which has been proven to be an effective strategy for the treatment of T2DM [5]. Meanwhile, excessive reactive oxygen species could attack cellular DNA, lipoproteins, and proteins, leading to dysregulation of cellular signaling and disruption of energy metabolism [6]. Many epidemiological studies have reported total antioxidant capacity may play an important role in reducing the risk of T2DM [7,8]. Recent studies have established that, as α-amylase inhibitors with excellent antioxidant activity, natural polyphenols could prevent or delay the onset of T2DM [9,10,11,12,13]. Therefore, dietary intervention with polyphenols may be a good strategy for improving diabetes or its complications.

The fruit of mangosteen (*Garcinia mangostana*) is a well-known tropical fruit widely cultivated in tropical Asia and Africa. It has a relatively small edible portion, i.e., only about 30% of the whole fruit; the pericarp which comprise 60–65% of mangosteen, and are usually discarded as waste after consumption [14,15]. Hence, complete processing of mangosteen pericarp is significant to the high-value utilization of mangosteen resources. Previous studies indicated that the pericarp contained the highest polyphenols levels among the different parts of mangosteen [16,17]. Meanwhile, it has been found that mangosteen pericarp extracts displayed strong antioxidant activity [18,19], anti-inflammatory activity [20,21], anti-adipogenic activity [22,23], therefore, resulting in potential therapeutic value in obesity, Alzheimer’s disease, cancer, among other diseases [24,25,26,27,28]. Therefore, there is a great demand for preparative enrichment of mangosteen pericarp polyphenols (MPPs) for in-depth pharmacological research and effective application in medical practice and dietary supplements of mangosteen pericarp. Hence, in this study, the extraction method of MPPs was first optimized, and the α-amylase inhibitory activity of MPPs was systematically evaluated both in vivo and in vitro.

## 2. Materials and Methods

### 2.1. Chemicals and Plant Materials

Mangosteen was purchased from a local supermarket in Hubei province. Its specimen was deposited in College of Food Science and Technology, Huazhong Agricultural University (voucher specimen number 2021-005). MS grade formic acid and methanol were purchased from Sigma-Aldrich (St. Louis, MO, USA). α-Amylase from porcine pancreas (16 U/mg) was purchased from Sigma-Aldrich (St. Louis, MO, USA). AB-8 macroporous resins were purchased from Nankaihecheng S&T Co., Ltd. (Tianjin, China). Gallic acid (purity ≥98%) and ascorbic acid were obtained from Shanghai Yuanye Co., Ltd. (Shanghai, China). All other reagents (BHT, HCl, etc.) were obtained from Sinopharm Chemical Reagent Factory (Shanghai, China).

### 2.2. MPPs Extraction

The pericarps of fresh mangosteen fruit were removed, and then washed with distilled water, dried naturally, and pulverized into fine powder. Mangosteen pericarp powders were reserved at −20 °C before extraction. Then, 3 g of mangosteen pericarp powders were extracted with hydrochloric acid aqueous with different solid to solvent ratios using a water bath. After that, the filtrate was collected by filtration under reduced pressure. The content of total polyphenols in filtrate was determined by the Folin phenol method and gallic acid was applied as the A standard [13]. The extraction yield was defined as the ratio of polyphenol mass (gallic acid equivalents) in filtrate to mangosteen pericarp powder mass.

#### Optimization of MPP Extraction Conditions

The influence of a single factor on the extraction yield was investigated using the solid to solvent ratios (1:35 g/mL, 1:40 g/mL, 1:45 g/mL, 1:50 g/mL, 1:55 g/mL, and 1:60 g/mL), pH values (0.5, 1.0, 1.5, 2.0, 2.5, and 3), extraction temperatures (50 °C, 60 °C, 70 °C, 80 °C, and 90 °C) and extraction times (0.5 h, 1.0 h, 1.5 h, 2.0 h, and 2.5 h) as variables, respectively. Based on the results of the single-factor experiment, a four-factor, three-level orthogonal experiment was designed (Appendix A).

### 2.3. Identification of Phenolic Compounds in MPPs

The purification of MPPs was achieved by dynamic adsorption and desorption on AB-8 macroporous resin column, followed by rotary evaporation to remove the ethanol. The final liquid was dried in a vacuum freeze dryer. A tentative analysis of the components of purified MPPs was performed with a Waters Acquity UPLC-QTOF-MS/MS system (Waters, Milford, MA) equipped with an Acquity UPLC BEH C18 column (2.1 mm × 100 mm, 1.7 μm) and UV detector, according to our previous method with minor modifications [29]. MPPs was dissolved in methanol, and analyzed at a flow rate of 0.4 mL/min. The UV detector condition was set as 280 nm. The injection volume was 1 μL, and mobile phase with solvent A (0.1% formic acid in water) and solvent B (acetonitrile). The gradient elution program was set up as follows: 0 min–0.5 min, 95% A and 5% B; 0.5 min–27 min, 95%–5% A and 5%–95% B; 27 min–29 min, 5%–95% A and 95%–5% B; 29 min–30 min, 95% A and 5% B. The mass spectrometry parameters were set as follows: capillary voltage 3.00 kV, cone gas flow rate 50 L/h, desolvation gas flow rate 600 L/h, emission source temperature 135 °C, desolvation temperature 350 °C, collision low energy 6.00 eV, collision high energy 20.00 eV–40.00 eV. The data were collected in negative mode and mass spectrometry scan mass range was 150 *m/z*–2000 *m/z*.

### 2.4. Total Antioxidant Properties of MPPs

The hydroxyl radical scavenging ability, superoxide radical scavenging ability, and DPPH· radical scavenging ability of MPPs were investigated based on the method of Sun et al. (2017) with slight modifications [30]. Ascorbic acid (VC) and 2,6-di-tert-butyl-4-methylphenol (BHT) were applied as the control.

### 2.5. Inhibition of Alpha-Amylase Activity In Vitro

#### 2.5.1. Alpha-Amylase Inhibitory Activity

The inhibitory potential of MPPs on α-amylase was determined as reported by Tan et al. (2017) with slight modifications [31]. MPPs solution (0.05 mg/mL–5.00 mg/mL) and 0.5 U/mL of α-amylase solution prepared in phosphate saline buffer (25 mmol/L, pH 6.8) were mixed and incubated at 37 °C for 10 min. After that, 0.5% starch solution was added and the reaction was incubated at 37 °C for 8 min. Then, 3,5-dinitrosalicylic acid (DNS) solution was added. The reaction solution was heated in boiling water for 5 min, and then cooled in an ice bath. The absorbance of the reaction solution was measured at 540 nm. Acarbose was used as a positive control. The experiment was divided into blank group, blank control group, sample group, and sample control group, corresponding to absorbance A_1_, A_0_, B_1_, and B_0_, respectively. The inhibition rate was calculated by the following formula:(1)$$\alpha -\mathrm{amylase}\;\mathrm{inhibition}\;(\%)=\frac{\left(A_{1}-A_{0}\right)-\left(B_{1}-B_{0}\right)}{A_{1}-A_{0}}\;\times \;100\%$$

#### 2.5.2. Alpha-Amylase Inhibition Kinetics

The concentration of α-amylase solution was 0.5 U/mL and the reaction rate was determined at different concentrations of MPPs (0 mg/mL, 1.0 mg/mL, and 4.0 mg/mL). The inverse of the reaction rate was used as the vertical coordinate and the reciprocal of the substrate concentration was used as the horizontal coordinate to make a Lineweaver–Burk plot to determine the type of inhibition of α-amylase by MPPs.

#### 2.5.3. Fluorescence Quenching

For fluorescence quenching, 1.0 mg/mL of amylase solution was mixed with 0.1 mg/mL, 0.2 mg/mL, 0.3 mg/mL, 0.4 mg/mL, 0.5 mg/mL, and 0.6 mg/mL MPPs solutions at a ratio of 1:1 separately and the mixtures were incubated for 15 min at 300 K or 310 K, respectively. The excitation wavelength was 280 nm and emission wavelengths were from 290 nm to 400 nm using a fluor spectrophotometer (Hitachi, Japan). To further analysis the type of fluorescence quenching type between the MPPs and α-amylase, the modified Stern-Volmer equation was used [32]:(2)$$\frac{F_{0}}{F}=\left(1+K\left[Q\right]\right)e^{\left[Q\right]VN/1000}$$
where *F_0_* and *F* are the fluorescence intensities in the absence and presence of MPPs, respectively; $\left[Q\right]$ is the concentration of the quencher and here it is the concentration of MPPs; *V* is the volume of the sphere; *N* is Avogadro’s constant. When *K*[*Q*] is small enough, then $1+K\left[Q\right]\approx e^{K\left[Q\right]}$, which is equivalent to $e^{\left[Q\right]VN}$. Thus, the equation becomes as follows:(3)$$\frac{F_{0}}{F}=e^{K_{FQ}\left[Q\right]}$$
where *K_FQ_* is the apparent static quenching constant. Since the molecular weights of MPPs is unknown, L·mg^−1^ and L·mg^−1^·s^−1^ were used as units of *K_FQ_* and *K_q_*, respectively, to detect the type of fluorescence quenching.

#### 2.5.4. Autodock

Molecular docking was accomplished using the AutoDock 4.2 software based on the method of Xie et al. [33]. The X-ray crystal structure of human pancreatic α-amylase (PDB ID: 1HNY) was downloaded from the Protein Data Bank and the waters and unique ligands were removed by PyMol. The 3D structures of typical compounds from MPPs were acquired from the PubChem database. Before autogrid, hydrogen bonds were added to macromolecule and ligand. Torsion bonds were detected in ligands. The grid box was set to 126 Å × 126 Å × 126 Å with grid spacing of 0.581 Å to include the whole enzyme. Fifty runs were created by using Lamarckian genetic algorithm searches. Finally, results from AutoDock were visualized and rendered by PyMol.

### 2.6. Inhibition of Alpha-Amylase Activity In Vivo

Male SD rats (200–220 g body mass) were purchased from the laboratory animal center of the Huazhong Agricultural University (Wuhan, China). All the experimental rats were housed under standard laboratory conditions at 22 ± 1 °C and 55 ± 10% humidity with 12 h of light/dark cycles and free access to diet and water. After one week of acclimatization, the rats (*n* = 24) were divided into four groups (control, acarbose, 2.5% MPPs, and 5% MPPs). As previous described with minor modifications, the rats after 12 h of fasting were administrated acarbose (30 mg/kg) or MPPs (20 mg/kg or 40 mg/kg), and the control group was administrated an equal amount of distilled water. Then, the maize starch solution that had been cooked in boiling water for 20 min was administrated orally to rats at a dose of 0.8 g starch/kg body weight [34]. Blood glucose was measured every 20 min using a Sinocare GA-3 blood glucose meter (Sanocare, China). All the experiments were performed in accordance with the Experimental Animal Review Committee of the Huazhong Agricultural University of China.

### 2.7. Statistical Analysis

The results were presented as mean ± SD (standard deviation). Comparisons between groups were analyzed by one-way ANOVA of SPSS 24 (IBM SPSS Statistics, IBM Corp., Armonk, NY, USA) using Duncan’s multiple-range test. The results were considered to be statistically significant when *p* < 0.05.

## 3. Results and Discussion

### 3.1. Extraction and Optimization Conditions of MPPs

Mangosteen pericarp has been found be a good source of polyphenol, and the polyphenols content was higher than that in rambutan peel, passion fruit peel, dragon fruit peel, and pineapple peel [35,36,37,38]. Therefore, the development and utilization of MPPs are of great significance for the efficient utilization of mangosteen pericarp. Thus, in this study, the extraction methods of polyphenols from the pericarp of mangosteen were first optimized. The influences of solid to solvent ratio, pH, temperature, and time on the yield of MPPs are shown in Appendix A. There was an overall increasing trend in the MPPs extraction yield in the range of 1:35 g/mL–1:60 g/mL (Appendix A). As there was no significant difference in the extraction yield in the range of 1:45 g/Ml–1:60 g/mL, therefore 1:45 g/mL was chosen as the optimum condition with an extraction rate of 2.93 ± 0.08%. For pH, MPPs extraction yield increased first, and then decreased slowly, when the pH was in the range of 0.5–3, and the maximum yield was 2.78 ± 0.09%. Therefore, the optimum pH was 1.5 (Appendix A). As shown in Appendix A, with an increase in temperature, the extraction yield of MPPs reached the highest at 70 °C, which was 1.42 ± 0.01% and after that the extraction yield decreased slowly. Therefore, an extraction temperature of 70 °C was selected. Meanwhile, the MPPs extraction was first raised, and then reduced ranging from 0.5 h to 2.5 h. When the extraction time was 1.5 h, the extraction yield reached a maximum of 1.53 ± 0.01%. Thus, the optimum extraction time was 1.5 h.

Based on the orthogonal array experiment results shown in Table 1, the factors influenced the extraction yield of MPPs in the following order: solid to solvent ratio > extraction temperature > pH > extraction time. The optimum conditions for acid hydrolysis were solid to solvent ratio of 1:50 g/mL, pH of 2, extraction temperature of 80 °C, and extraction time of 2 h; the best extraction yield was 3.09 ± 0.077%. This yield was close to that of Mohammed et al. (2019) with microwave-assisted ethanol extraction, and was higher than Muzykiewicz et al. (2020) with ultrasound-assisted extraction and Sungpud et al. (2020) with virgin coconut oil (VCO) extraction [17,39,40].

### 3.2. Tentative Identification of MPPs

The polyphenol content was 86.08 ± 2.39% in the purified MPP extraction. Then, the tentative identification of purified MPPs was performed by UPLC-ESI-QTOF-MS/MS. The UPLC chromatogram is shown in Figure 1 and Table 2. A total of 21 major compounds were identified by UPLC-ESI-QTOF-MS/MS preliminarily as compared with published data [41,42,43,44,45,46]. Data showed that there were 18 polyphenol compounds in MPPs, and there were 7 compounds identified as characteristic polyphenols from mangosteen pericarp such as compounds 1 (β-mangostin), 17 (garcimangosxanthone C), 18 (garcinone C), and 20 (garcinone D). However, α-mangostin and other representative xanthones were not observed in this study, partly due to the extract method [47].

### 3.3. Antioxidant Properties of MPPs

Free radicals are highly reactive molecules produced in an organism during cellular respiration and normal physiological metabolism which are closely associated with a number of physiological and pathological processes in the body [48,49,50]. A previous study found that oxidative stress and free radicals production were associated with T2DM and its complications [51]. Meanwhile, excessive reactive oxygen species could attack cellular DNA, lipoproteins, and proteins, leading to cellular oxidative damage [6]. Many epidemiological studies have reported antioxidants may play an important role in reducing the risk of T2DM [7,8]. In this study, MPPs exhibited considerable hydroxyl radical, superoxide radical, and DPPH**·** radical scavenging activities. As the results showed in Table 3, MPPs exhibited strong antioxidant ability; the IC_50_ values for the scavenging hydroxyl radical, superoxide radical, and DPPH**·**; were 2.24 ± 0.10 mg/mL, 1.47 ± 0.11 mg/mL, and 0.15 ± 0.004 mg/mL, respectively.

The antioxidant capacity of phenolics was mainly attributed to the specific aromatic nucleus and highly conjugated system of multiple hydroxyl groups, those phenolic compounds which possess beneficial hydrogen or electron atom donors that have the ability to scavenge free radicals and reactive oxygen compounds [52]. For instance, the presence of two phenolic hydroxyl groups on the benzene rings of compounds 1 (β-mangostin), 18 (garcinone C), and 20 (garcinone D) were potent electron donors, which was consistent with their effective antioxidant effects [30,42].

### 3.4. Inhibition of Alpha-Amylase Activity In Vitro and Possible Mechanism

Starch digestion is an important target for controlling blood glucose. It has been reported that postprandial hyperglycemia may be alleviated by polyphenols by inhibiting the activities of starch-hydrolyzing enzymes such as α-amylase [9,31,53]. In this study, the interaction and possible mechanism between MPPs and α-amylase were investigated preliminarily. The inhibition of α-amylase by MPPs was increased as the concentration (0–0.5 mg/mL) increased (Figure 2A). However, when the polyphenol concentration was greater than 1.5 mg/mL, the inhibition gradually steadied at around 84.63%. The IC_50_ value for the inhibition of α-amylase by MPPs was calculated to be 0.28 mg/mL and by acarbose was 0.05 mg/mL. As the Lineweaver–Burk plots of MPPs on the inhibition of α-amylase show in Figure 2B, the three fitted lines in the image correspond to the results at 0 mg/mL, 1.0 mg/mL, and 4.0 mg/mL of MPPs solution, respectively, and intersect at a point on the horizontal axis at coordinates (−0.05, 0) approximately. As compared with the line of α-amylase alone, the *K_m_* of the enzymatic reaction catalyzed by α-amylase was essentially unchanged and *V*′*_max_* decreased when MPPs were present, where *V*′*_max_* was the apparent maximum rate (apparent *V_max_*). In summary, the type of inhibition of α-amylase by MPPs was determined to be non-competitive.

There are growing numbers of studies on the beneficial effects of polyphenols extracted from mangosteen. For instance, the extract of mangosteen, most of which were xanthones, showed effective α-glucosidase inhibitory activity with an IC_50_ value of 3.2 μg/mL and γ-mangostin, as a pure compound in the extract, exhibited the most potent activity and mixed type of inhibition [54]. In addition, 3-isomangostin from the dichloromethane extract of mangosteen demonstrated powerful inhibitory activity for aldose reductase (IC_50_ value of 3.48 μM) [55]. Moreover, enzyme activity assays have proven that garcinone E had effectual inhibition for protein tyrosine phosphatase 1B (PTP1B), a valid target for T2DM drugs, with an IC_50_ value of 0.43 μM [25].

The intrinsic fluorescence of proteins or peptides has been derived from tryptophan (Trp), tyrosine (Tyr), and phenylalanine (Phe) residues, whereas the intrinsic fluorescence of α-amylase has mainly been contributed by Trp residues [56]. Accordingly, fluorescence intensity was used, here, to analyze the binding mechanism of the interaction between MPPs and α-amylase. Figure 2C,D shows the course of the fluorescence quenching spectra of α-amylase with different concentrations of MPPs at 300 K and 310 K. The intrinsic fluorescence intensity of α-amylase decreased sharply as the MPP concentration increased (29.16% at a MPP concentration of 0.05 mg/mL). The results suggested that MPPs could interact with α-amylase and quench the intrinsic fluorescence of α-amylase.

Fluorescence quenching mechanisms are usually classified as three types, namely, static quenching (the formation of a complex without fluorescence formed between the fluorophore and the quencher), dynamic quenching (the collision of the fluorophore with the quencher that causes the fluorophore in the excited state to lose energy and return to the ground state), and a mixed type of both [57]. The plots of Stern-Volmer were used to analyze the quenching type. The linear plots generally suggested only one class of fluorophores in macromolecule and all of them could be captured by quenchers, indicating that only static quenching or dynamic quenching occurred [32]. The fluorescence quenching mechanism between gallic acid and α-amylase was a static quenching type, which meant that a complex was formed between gallic acid and α-amylase [58]. Similarly, there was a steady complex formed, leading to non-radiation energy transferring and static fluorescence quenching of α-amylase in the present of quercetin, isoquercetin, and rutin [56].

In the case of an upward curvature in the plot, both quenching types may exist with the same quencher or existing apparent static quenching characterized by the “sphere of action” model [59]. Figure 2E shows the Stern-Volmer plots of fluorescence quenching of pancreatic α-amylase by MPPs at different temperatures (300 K and 310 K), presenting an upward curvature and concave toward the y axis. This was consistent with the plot from a previous study of the interaction between mangosteen and bovine serum albumin (BSA) [60]. Hence, the linear model was no longer applicable and, in this study, the modified Stern-Volmer equation was used to calculate the quenching constant. Table 4 summarizes the constants calculated from the modified equation, which suggest that the modified model is suitable for studying the binding mechanism between MPPs and α-amylase (R^2^ = 0.9960 (300 K) and 0.9839 (310 K)). The values of *K_FQ_* obtained from the exponential fit method were 8.781 mL·mg^−1^ (300 K) and 8.503 L·g^−1^(310 K), which revealed the high affinity of MPPs for α-amylase (Table 4). MPPs in our study showed greater affinity than young apple polyphenols and black tea extracts due to their higher apparent static quenching constant [59,61]. It was presumably that the high binding affinity of MPPs for α-amylase was due to the wide range of polyphenols they contained, most of which could act as a quencher through different mechanisms. Based on the UPLC-ESI-QTOF-MS/MS data, (E)C (compound 6) was an abundant constituent in the MPPs comparatively and it was confirmed in another study that it could exhibit fluorescence quenching activity, but the binding capacity was not as good as MPPs [59]. Cyanidin-3-*O*-glucoside (compounds 4 and 13) and pelargonidin-3-*O*-glucoside (compound 19) showed fluorescence quenching activity by static mode [62]. Thus, MPPs may cause changes in the microenvironment of Trp residues of α-amylase due to hydrogen bonds and hydrophobic interaction in a dose-dependent manner [59].

Based on the preliminary results of the mass spectra, the main components of MPPs could be divided into the characteristic components of mangosteen pericarp, i.e., anthocyanins and proanthocyanidins, therefore, the typical components of these were selected to explore the interaction with α-amylase using AutoDock Tools and visualized by PyMol. As shown in Figure 3B–H, the binding sites of these compounds were adjacent to the active sites of α-amylase (PDB ID: 1HNY), containing Trp59, Asp197, Asp300, Arg303, Glu233, and His305 [63]. Previous studies have confirmed that Glu233 of α-amylase played a key role in amylase catalysis and the amylase activity could show a 10^3^-fold decrease when the side chain group of Glu233 was substituted [64]. As shown in Figure 3, most polyphenols found in MPPs could form hydrogen bonds with Glu233 or His 315. Among these docking results, garcimangosone D (compound 9) showed the strongest activity and could form hydrogen bonds with Trp59, Asp197, Glu233, and His305 of α-amylase (Figure 3B). Proanthocyanidin A2 (compound 12) and (E)C (compound 6) showed stronger inhibition activity due to lower binding energy and root-mean-square deviation (RMSD) (Appendix A). These results indicated that MPPs could directly bind to the residues near the active site through hydrogen bond interaction and, thus, inhibit the activity of α-amylase.

### 3.5. Postprandial Blood Glucose Response to Starch with MPPs

From the above in vitro results, MPPs showed the considerable inhibitory ability of α-amylase, suggesting that MPPs may have the potential to inhibit starch digestion in vivo. Therefore, in this study, the effects of MPPs on postprandial blood glucose levels were also tested in rats. Figure 4A shows the levels of blood glucose in rats after the acute administration of maize starch with different concentrations of MPPs. Glucose levels in the control group peaked at 20 min after ingestion of the starch solutions, then declined, and eventually reaching baseline after 80 min. The peak of blood glucose response and the area under the curve (AUC) of blood glucose were both significantly lower in rats treated with MPPs as compared with the control group (Figure 4A,B), but the difference between high and low dose MPPs groups on postprandial blood glucose was not significant. Results from acute in vivo experiments showed that MPPs could reduce postprandial blood glucose, with effects similar to acarbose.

Globally, diabetes mellitus is among the top 10 causes of death and is one of the most critical public health problems. Long-term hyperglycemia and insulin resistance will cause many organ damages and life-threatening complications such as nephropathy, retinopathy, diabetic foot ulcer, neuropathy, cardiovascular, and cerebrovascular disease [1].

Until now, modification of dietary habits and the use of drugs that prevent the digestion of starch seem to be the simplest modifiable and cost-effective methods. There is growing evidence showing that people with diets rich in foods containing a high content of polyphenols and high antioxidant capacity compounds may have a lower risk of T2DM [4,10,11]. Polyphenols have been demonstrated to be a considerable postprandial glycemic control in many experiments *ro*. It was found that postprandial glycemic responses and insulin levels in rats were inhibited by young apple polyphenols by about 10% as compared with rats only fed with starch [9]. Glucagon-like peptide 1 (GLP-1) secretion also tended to be increased after consumption of polyphenol extracts from coffee as compared with a placebo [65]. Sodium-dependent glucose transporter (SGLT1) and glucose transporter 2 (GLUT2) have been confirmed to have a physiological link to the expression of incretin, control of which is one of the pathways to regulate blood glucose homeostasis in vivo [66]. Polyphenol extracts from apple and blackcurrant decreased postprandial glucose and glucose-dependent insulinotropic polypeptide (GIP) levels after a high-carbohydrate meal in healthy participants, which may be explained by the inhibition of intestinal GLUT2 and SGLT1 [66].

The preventive effects of dietary polyphenols on T2DM can be summarized as: inhibition of α-amylase, α-glucosidase, or aldose reductase activity; protection of pancreatic cells from glucose; antioxidant effect; reduction in inflammatory stress, inhibition of the formation of advanced glycation end-products (AGEs); influence on other food ingredient bioaccessibility, etc. [5,67,68]. Among them, inhibiting the digestion of starch and the adsorption of glucose may play the key roles for the anti-diabetes potential of polyphenols. Tea polyphenols have been proven to reduce the catalytic activity of porcine pancreatic α-amylase [69]. Chlorogenic acid has been confirmed to modify the secondary structure of porcine pancreatic α-amylase mainly by interacting with amino acid residues around the active site of α-amylase through hydrogen bond interactions [70]. Similar results were observed in this study, i.e., MPPs formed hydrogen bonds with amino acid residue Glu233, a vital catalytic site of α-amylase, and consequently exhibited the inhibition ability on α-amylase. In addition, it has also been shown that α-glucosidase, an essential enzyme in hydrolyzing carbohydrates, could be inhibited by young apple polyphenols and mung bean skin bound polyphenols [9,13]. In addition to the above enzyme inhibition, polyphenols from acorn leaves have been shown to be capable of increasing the viability of pancreatic beta cells [71]. Umadevi et al.(2014) found that gallic acid could attenuate AGE formation, decreasing the risk of T2DM [72].

This study indicated that polyphenols from mangosteen could reduce postprandial blood glucose by inhibiting α-amylase activity. In other studies, β-mangostin (compound 1) was demonstrated to have comparative anti-inflammatory activity and aldose reductase inhibition [55,73,74]. Abdallah et al. (2016) found four bioactive constituents that had AGE inhibition activity from the methanol extract of mangosteen including garcimangosone D (compound 9) and (E)C (compound 6), whereas (E)C showed the strongest potential [75]. Moreover, the other proposed beneficial effects of (E)C can be summarized as capacities to increase satiety, ameliorate insulin resistance, improve insulin sensitivity, mitigate oxidative stress, and downregulate expression of inflammatory factors and related pathways [76,77,78,79]. Proanthocyanidin A2 (compound 12) has also shown effective α-glucosidase inhibition with an IC_50_ value of 1.99 μg/mL [46]. In addition, anthocyanin, a relatively abundant component in MPPs, and cyanidin-3-*O*-glucoside (compounds 4 and 13), a widely distributed anthocyanin, had the effect to alleviate insulin resistance induced by palmitic acid at a molecular level [80]. Taken together, these finding indicate that MPPs may help to ameliorate postprandial hyperglycemia with different underlying mechanisms.

## 4. Conclusions

An efficient extraction method for enrichment of MPPs was first developed, and the IC_50_ value of MPPs for α-amylase was 0.28 mg/mL. In addition, MPPs significantly reduced the blood glucose peak at 20 min and the AUC in rats after ingestion of starch solutions. MPPs could alter the natural structure of the α-amylase, resulting in inhibitory activity on α-amylase. Taken together, MPPs may have the potential to develop as a functional supplement for blood glucose control and T2DM prevention. These results demonstrate that the pericarp of mangosteen fruit is a potential source of dietary polyphenols that can be used as natural-source food supplements with health benefits, which could promote the efficient utilization of mangosteen fruit.

## Figures and Tables

**Figure 1 foods-11-01001-f001:**
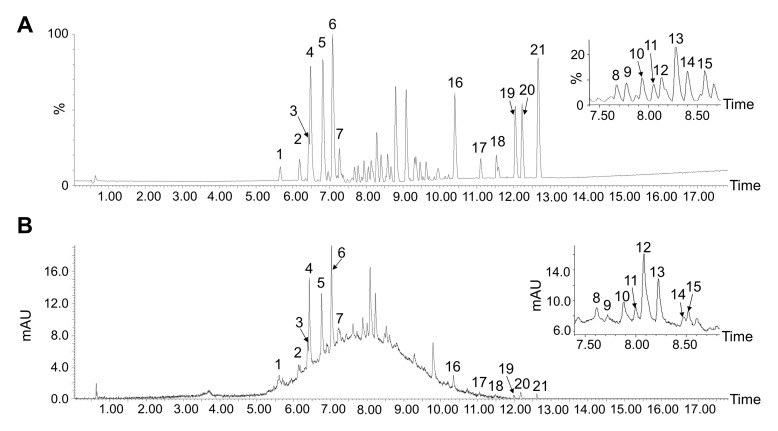
(**A**) BPI chromatogram of MPPs in negative mode; (**B**) UV chromatogram of MPPs at 280 nm.

**Figure 2 foods-11-01001-f002:**
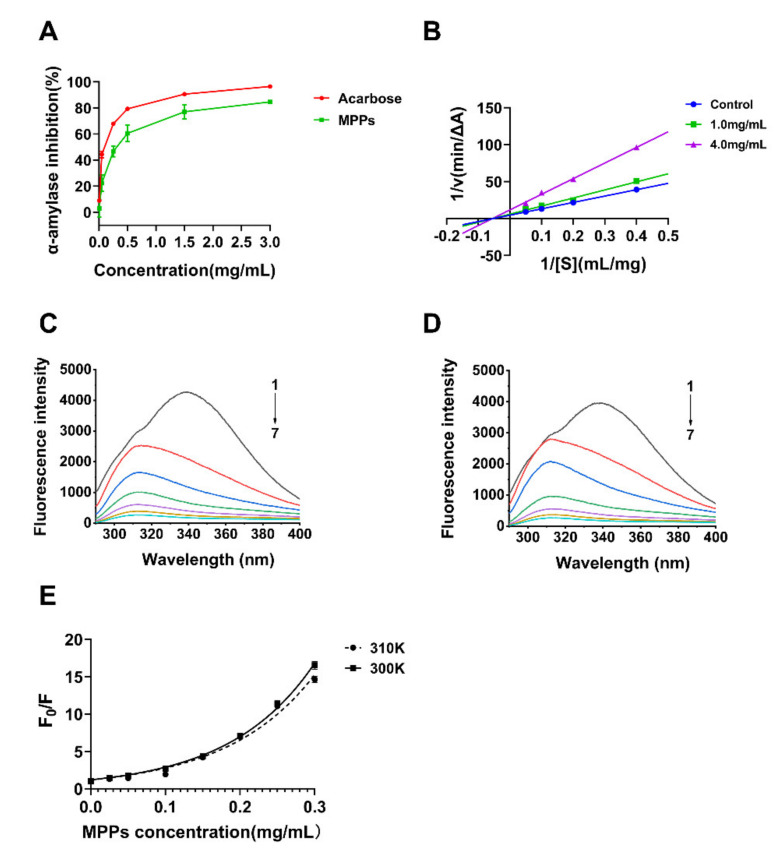
Inhibition of α-amylase by MPPs in vitro: (**A**) Inhibitory effect of MPPs on the α-amylase activity; (**B**) Lineweaver-Burk plots of MPPs on α-amylase; (**C**,**D**) fluorescence spectra of α-amylase with various amounts of MPPs, (C) T = 300 K and (D) T = 310 K. [α-amylase] = 0.50 mg/mL; [MPPs] = 0 mg/mL (1), 0.05 mg/mL (2), 0.10 mg/mL (3), 0.15 mg/mL (4), 0.20 mg/mL (5), 0.25 mg/mL (6), and 0.30 mg/mL (7), respectively; (**E**) The Stern-Volmer plots of α-amylase quenched by MPPs at different temperatures (300 K and 310 K).

**Figure 3 foods-11-01001-f003:**
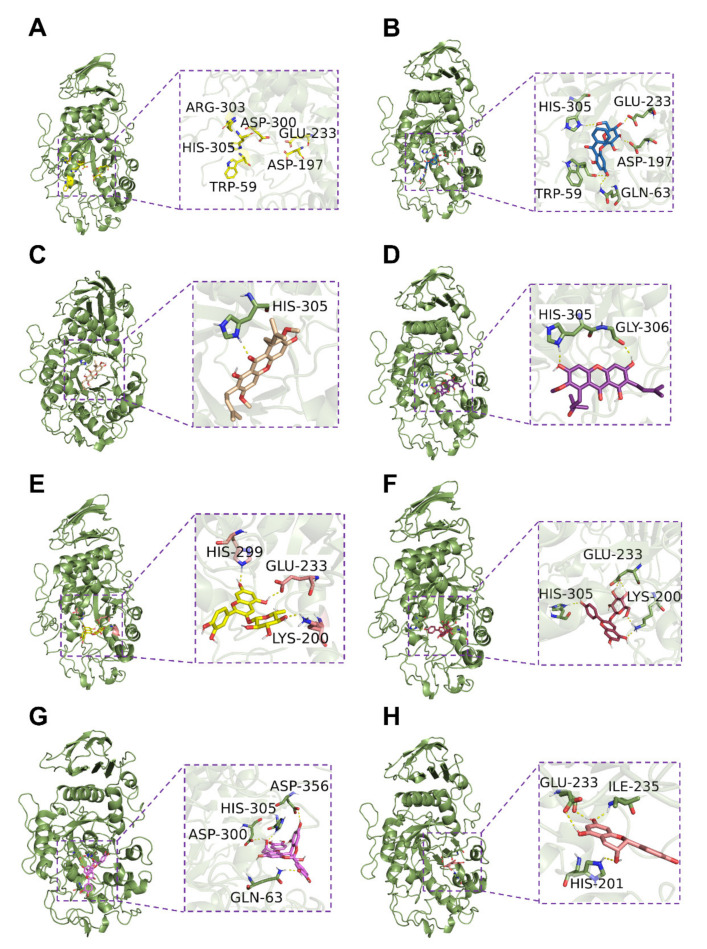
Predicted binding mechanism of α-amylase with representative compounds in MPPs: (**A**) Active sites of human pancreatic α-amylase (PDB ID: 1HNY); (**B**) garcimangosone D; (**C**) β-mangostin; (**D**) garcinone D; (**E**) cyanidin-3-*O*-glucoside; (**F**) pelargonidin-3-*O*-glucoside; (**G**) proanthocyanidin A2; (**H**) (**E**) C.

**Figure 4 foods-11-01001-f004:**
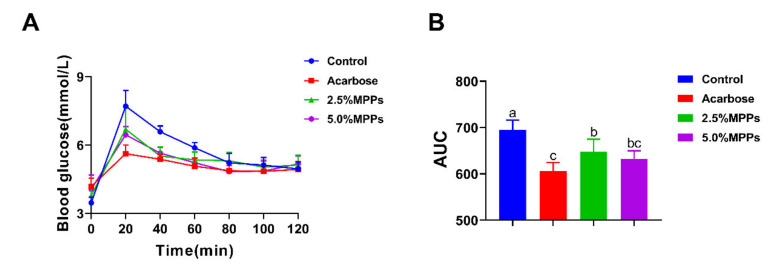
Postprandial change of blood parameters in rats with the presence of MPPs: (**A**) Blood glucose in rats treated with MPPs (*n* = 6); (**B**) AUC of blood glucose in rats treated with MPPs (*n* = 6). Data were analyzed by ANOVA and Duncan’s multiple-range test. Different letters mean significant differences (*p* < 0.05).

**Table 1 foods-11-01001-t001:** Orthogonal test results of extraction yield of MPPs.

Number	Mangosteen Pericarp Powder/g	Solid to Solvent Ratio (A)	pH (B)	Temperature (C)	Time (D)	Extraction Yield (%)
1	3.00	1	1	1	1	2.65 ± 0.07
2	3.00	1	2	2	2	2.77 ± 0.09
3	3.00	1	3	3	3	2.98 ± 0.05
4	3.00	2	1	2	3	2.91 ± 0.04
5	3.00	2	2	3	1	3.07 ± 0.08
6	3.00	2	3	1	2	2.85 ± 0.05
7	3.00	3	1	3	2	3.08 ± 0.07
8	3.00	3	2	1	3	2.99 ± 0.06
9	3.00	3	3	2	1	3.09 ± 0.08
K_1_		8.40	8.64	8.49	8.81	
K_2_		8.83	8.83	8.77	8.70	
K_3_		9.16	8.92	9.13	8.88	
k_1_		2.80	2.88	2.83	2.94	
k_2_		2.94	2.94	2.92	2.90	
k_3_		3.05	2.97	3.04	2.96	
R		0.25	0.09	0.21	0.06	
Order	A > C > B > D					
Optimum Levels		A_3_	B_3_	C_3_	D_3_	
Optimum Factors	A_3_B_3_C_3_D_3_					

**Table 2 foods-11-01001-t002:** Tentative identification by UPLC-ESI-QTOF-MS/MS of MPPs.

No.	Retention Time	Absorption Peak Wavelength (nm)	Precursor Ion [M-H]^−^(*m/z*)	Main Fragment Ions (*m/z)*	Proposed Molecular Formula	Tentative Identification
1	5.66	225	423.0916	423.0912	C_25_H_28_O_6_	β-Mangostin
2	6.18	278	577.1353	577.1341 451.1021 425.0861 407.0760 289.0695	C_30_H_26_O_12_	B-type (E)C dimer
3	6.44	280	523.1481	523.1480 331.0799 289.0659		Unknown
4	6.48	288	449.1087	449.1073 359.0753 329.0648 301.0659	C_21_H_21_O_11_^+^	Cyanidin-3-*O*-glucoside
5	6.82	279	577.1353	577.1341 451.1021 425.0861 407.0760 289.0695	C_30_H_26_O_12_	B-type (E)C dimer
6	7.10	280	289.0705	289.0694 245.0792	C_15_H_14_O_6_	(E)C
7	7.27	280	865.2007	865.1994 577.1340 407.0758 289.0695	C_45_H_38_O_18_	B-type (E)C trimer
8	7.68	279	577.1353	577.1352 449.0882 407.0765 289.0701	C_30_H_26_O_12_	B-type (E)C dimer
9	7.77	238	391.1021	391.1013 289.0699 229.0484	C_19_H_20_O_9_	Garcimangosone D
10	7.93	280	863.1870	863.1870 575.119 289.0697	C_45_H_36_O_18_	A-type (E)C trimer
11	8.03	278	275.0540	275.0542 243.0275	C_14_H_12_O_6_	4,6,3′,4′-Tetrahydroxy-2-methoxybenzophenone
12	8.14	285	575.1205	575.1189 449.0891 407.0762 285.0383	C_30_H_24_O_12_	Proanthocyanidin A2
13	8.28	523	449.1080	449.1064 303.00487 285.0383 151.0014	C_21_H_21_O_11_^+^	Cyanidin-3-*O*-glucoside
14	8.40	540	610.4193	610.4177 564.4135 289.0701	C_27_H_31_O_16_^+^	Cyanidin-3-*O*-sophoroside
15	8.57	239	505.1376	505.1376 449.0899 381.1173 241.0000	C_21_H_30_O_12_S	4-*O*-sulpho-β-D-glucopyranosyl abscisate
16	10.42		443.1698	443.1703 428.1467 369.0966 297.0387		Unknown
17	11.10	249	345.0958	345.0958 272.0303 243.0274	C_19_H_22_O_6_	Garcimangosxanthone C
18	11.54	250	413.1585	413.1584 357.0958 329.1005	C_23_H_26_O_7_	Garcinone C
19	12.05	320	431.1708	431.1689 358.1031	C_21_H_21_O_10_^+^	Pelargonidin-3-*O*-glucoside
20	12.23	306	427.1754	427.1754 353.1013 297.0386	C_24_H_28_O_7_	Garcinone D
21	12.67	317	445.1861	445.1857 371.1123 357.0964		Unknown

**Table 3 foods-11-01001-t003:** The antioxidant index IC_50_ values of MPPs, VC and BHT.

Antioxidant	Hydroxyl Radical	Superoxide Radical	DPPH
MPPs (mg/mL)	2.24 ± 0.10 ^a^	1.47 ± 0.11 ^b^	0.15 ± 0.004 ^b^
VC * (mg/mL)	1.93 ± 0.01 ^b^	0.17 ± 0.003 ^c^	0.11 ± 0.01 ^b^
BHT * (mg/mL)	0.83 ± 0.02 ^c^	3.60 ± 0.47 ^a^	1.15 ± 0.18 ^a^

* VC, ascorbic acid; BHT, 2,6-di-tert-butyl-4-methylphenol. Data were showed as mean ± SD. Different letters indicated significant differences (*p* < 0.05).

**Table 4 foods-11-01001-t004:** The Stern-Volmer regression equations for the fluorescence quenching of α-amylase by MPPs at different temperatures.

	T/K	Regression Equation	R^2^	K_FQ_ (L·g^−1^)	K_q_ (10^8^L·g^−1^·s^−1^)
MPPs + α-amylase	300	y = 1.207 exp (8.781x)	0.9960	8.781	2.927
MPPs + α-amylase	310	y = 1.193 exp (8.503x)	0.9839	8.503	2.834

## Data Availability

The data presented in this study are available on request from the corresponding.

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
