# Peer review of "Inhibitory Effects against Alpha-Amylase of an Enriched Polyphenol Extract from Pericarp of Mangosteen (Garcinia mangostana)"

_foods, 2022, doi:10.3390/foods11071001_

Round 1

Reviewer 1 Report

The paper describes a process of enriching an extract of pericarp of mangosteen, determination of its composition and inhibitory effects against a-amylase. The paper has areas of improvement, which I have indicated on scribbles of the attached. These include:

  1. Addressing a flaw on references, where in-text are numbered whereas the list at the end is in alphabetical order
  2. I suggest amending the title to be readable:- Inhibitory effects against a-amylase of an enriched polyphenol extract from pericarp of Mangosteen, Garcinia mangostana
  3. Address the other provided comments on the attached. 

Author Response

Thank you very much for reviewing our manuscript and also giving us excellent suggestions and comments. As your comments, we carefully checked the manuscript, and the references has been revised as the requirement of this journal, and the title also has been revised as “Inhibitory Effects against a-amylase of an Enriched Polyphenol Extract from Pericarp of Mangosteen (Garcinia mangostana)”. In addition, we carefully checked the manuscript, and also many detail mistakes have been revised, please kindly find them in the revised manuscript.

Reviewer 2 Report

The study underscores the importance of the polyphenolic extract of Mangosteen (Garcinia mangostana L.) Pericarp in ameliorating type 2 diabetes mellitus (T2DM) via appreciable inhibition of α–amylase in vitro, in vivo, and in silico. Easy accessibility to the plant by locals and the non-toxic nature of the polyphenolic extract make this study pertinent and beneficial. and it can be accepted for publication after taking the following minor revisions into consideration.

I would like the authors to fill the following gaps:

  • In what form is the plant extract consumed by the locals. i.e what medium of extraction do they employ for consumption except for eating it as it is?
  • Is there any associated toxicity of consuming this fruit?
  • Can the authors add a voucher specimen number for the plant if deposited?
  • Some subtitles are not numbered please make sure to add the appropriate numbering.
  • The sections Results and Discussion are mostly well written, informative, and interesting. More new references should be included, especially in the part dealing with the antioxidants activities and the inhibition of α-amylase. In MDPI journals, such as Foods, Plants, and Biomolecules, several papers have been published on this topic.
  • Please separate the conclusion section from the results and discussion part
  • Please you have to number and reformat the references section to conform to the Foods referencing style.

Finally, I'd like to point out that the study was quite revealing. Further studies to unravel and elucidate the isolated pharmacological active factors contained in the plant, which might be responsible for the anti-diabetic properties of the plant, will be highly beneficial.

Author Response

  1. The study underscores the importance of the polyphenolic extract of Mangosteen (Garcinia mangostana L.) Pericarp in ameliorating type 2 diabetes mellitus (T2DM) via appreciable inhibition of α–amylase in vitro, in vivo, and in silico. Easy accessibility to the plant by locals and the non-toxic nature of the polyphenolic extract make this study pertinent and beneficial. and it can be accepted for publication after taking the following minor revisions into consideration.

Thank you very much for reviewing our manuscript and also giving us excellent suggestions and comments. As your comments, we carefully checked the manuscript, and our point-by-point responses for the comments are as follows:

  1. In what form is the plant extract consumed by the locals. i.e what medium of extraction do they employ for consumption except for eating it as it is?

As a Traditional Thai medicine, the mangosteen pericarp commonly used to treat abdominal pain, diarrhea, dysentery, infectious trauma, and so on. Water was the main extraction medium.

  1. Is there any associated toxicity of consuming this fruit?

As a popular fruit, the flesh of the mangosteen has no toxicity, and there were no useful data about the toxicity of the extraction of the pericarp, this needs to further investigate.

  1. Can the authors add a voucher specimen number for the plant if deposited?

Thank you very much for your suggestions. Now we have added this information in the revised manuscript. (L65, P2)

  1. Some subtitles are not numbered please make sure to add the appropriate numbering.

Thank you very much for your suggestions. Now we have added this information in the revised manuscript.

  1. The sections Results and Discussion are mostly well written, informative, and interesting. More new references should be included, especially in the part dealing with the antioxidants activities and the inhibition of α-amylase. In MDPI journals, such as Foods, Plants, and Biomolecules, several papers have been published on this topic.

Thank you very much for your suggestions. Now we have added some references in the revised manuscript.

  1. Please separate the conclusion section from the results and discussion part.

Thank you very much for your suggestions. Now we have added this information in the revised manuscript.

  1. Please you have to number and reformat the references section to conform to the Foods referencing style.

Thank you very much for your suggestions. Now we have added this information in the revised manuscript.

  1. Finally, I'd like to point out that the study was quite revealing. Further studies to unravel and elucidate the isolated pharmacological active factors contained in the plant, which might be responsible for the anti-diabetic properties of the plant, will be highly beneficial.

Thank you very much for your suggestions. We actually agreed with you that many study are needed to further investigate the isolated pharmacological active factors, and also the underlying mechanism.

Reviewer 3 Report

The article entitled "Inhibitory Effect of Polyphenol Extracts from Mangosteen (Garcinia mangostana) Pericarp on α-Amylase" reported phenolic characterization of this tropical fruit and its enzyme inhibition activity; The experimental methods were widespread, their results could be helpful to promote the use of the pericarp of this fruit as a functional additive in treatments on MD2. However,  the writing and discussion of the results should be improved.

Below please find my suggestions and comments:

- It is recommended to rewrite the study's objectives to improve reading
- It is recommended to add a picture of the fruit.
- Was the sample treated as an infusion or by decoction? Please clarify and include in the method.
- Improve table S2. Table S2 provides more relevant information to the reader than Figure 1. Consider a replacement. 
- In the identification of MPP compounds, did you identify any new compounds?

- It is recommended to add a scheme with the most important structures.
- The paragraph of lines 221 to 233 must be included in the introduction
- The discussion in section 3.3 needs to be improved
- It is recommended to delete figure 3.
- Include in table 2 the abbreviations of the standards (e.g., BHT: Butylhydroxytoluene). VC is ascorbic acid?
- Improve visual supports throughout the manuscript (Figures, Tables, Scheme).
- Conclusions should be rewritten - the do not repeat information given in the discussion is advisable. 
- It is recommended not to use "it was presumed" and change to "it was determined." Review entire section 3
- Section 3 should be Discussion and Results. Also, add section 4, Conclusion.

Lines 33-34: it is recommended to change "There were many factors that influence" to "Many factors influenced"
Lines 34-35: it is recommended to improve the sentence
Line 39: Change "have" and "to" to "has" and "for"
Line 43: Delete "and"
Line 52: Change "and so on" to "among other diseases"
Line 54: Add "s" to "supplementS"
Line 55: add "the α-amylase"
Line 177: Add "the development"
Line 179: Add "the pericarp"
Lines 195-197: It is recommended to rewrite the sentence. The extraction yield is expressed in % and not in mg GAE/g.
Line 198: add "of Mohammed"
Lines 206-207: For greater clarity to the reader, it is recommended to rewrite this sentence.
Line 209: add "A total"
Line 229-230: it is recommended to change "In this study, MPPs exhibited considerable hydroxyl radical, superoxide radical, DPPH• radical scavenging activities" to "In this study, MPPs exhibited considerable hydroxyl, superoxide, and DPPH• radical scavenging activities"
Line 289: Add "s" to "mechanismS"
Line 302 Add "In the case"
Line 307: add "a previous"; "the interaction"
Line 309: add "the quenching"
Line 312: add "the exponential"
Line 320: choose between "another study" or "other studies"
Line 323: add "in the microenvironment"
Line 332: add "are shown"
Line 351: add "the considerable"
Line 362: delete "those of"
Lines 377-381: it is recommended to improve the sentence
Line 406: change "In this study, the results" to "This study"

Author Response

- It is recommended to rewrite the study's objectives to improve reading

Thank you very much for your suggestions. Now we have revised it in the revised manuscript.

- It is recommended to add a picture of the fruit.

Thank you very much for your suggestions. We actually agreed with you, however, for the reason that this period is not the mature season of mangosteen, and also COVID-19, it was difficult to obtain fresh mangosteen.

- Was the sample treated as an infusion or by decoction? Please clarify and include in the method.

Thank you very much for your comment. The sample was extract using water within different pH using a water bath. We have added the detail information in the manuscript.

- Improve table S2. Table S2 provides more relevant information to the reader than Figure 1. Consider a replacement.

Thank you very much for your suggestions. Now we have revised it in the revised manuscript.

- In the identification of MPP compounds, did you identify any new compounds? It is recommended to add a scheme with the most important structures.

Thank you very much for your suggestions. As we all know that UPLC-MS method could only obtain a tentative identification, therefore, in this study, we did not identify any new compounds, and we will also isolate different component from the pericarp of mangosteen and investigate the inhibitory ability of α-amylase.

- The paragraph of lines 221 to 233 must be included in the introduction

Thank you very much for your suggestions. Now we have revised it in the revised manuscript.

- The discussion in section 3.3 needs to be improved

Thank you very much for your suggestions. Now we have revised it in the revised manuscript.

- It is recommended to delete figure 3.

Thank you very much for your suggestions. Now we have deleted figure 3 in the revised manuscript.

- Include in table 2 the abbreviations of the standards (e.g., BHT: Butylhydroxytoluene). VC is ascorbic acid?

Thank you very much for your suggestions. Now we have added this information in the revised manuscript.

- Improve visual supports throughout the manuscript (Figures, Tables, Scheme).

Thank you very much for your suggestions. Now we have revised it in the revised manuscript.

- Conclusions should be rewritten - the do not repeat information given in the discussion is advisable.

Thank you very much for your suggestions. Now we have revised it in the revised manuscript.

- It is recommended not to use "it was presumed" and change to "it was determined." Review entire section 3

Thank you very much for your suggestions. Now we have revised it in the revised manuscript.

- Section 3 should be Discussion and Results. Also, add section 4, Conclusion.

Thank you very much for your suggestions. Now we have revised it in the revised manuscript.

- Lines 33-34: it is recommended to change "There were many factors that influence" to "Many factors influenced"

Thank you very much for your suggestions. Now we have revised it in the revised manuscript.

- Lines 34-35: it is recommended to improve the sentence

Thank you very much for your suggestions. Now we have revised it in the revised manuscript.

- Line 39: Change "have" and "to" to "has" and "for"

Thank you very much for your suggestions. Now we have revised it in the revised manuscript.

- Line 43: Delete "and"

Thank you very much for your suggestions. Now we have revised it in the revised manuscript.

- Line 52: Change "and so on" to "among other diseases"

Thank you very much for your suggestions. Now we have revised it in the revised manuscript.

- Line 54: Add "s" to "supplementS"

Thank you very much for your suggestions. Now we have revised it in the revised manuscript.

- Line 55: add "the α-amylase"

Thank you very much for your suggestions. Now we have revised it in the revised manuscript.

- Line 177: Add "the development"

Thank you very much for your suggestions. Now we have revised it in the revised manuscript.

- Line 179: Add "the pericarp"

Thank you very much for your suggestions. Now we have revised it in the revised manuscript.

- Lines 195-197: It is recommended to rewrite the sentence. The extraction yield is expressed in % and not in mg GAE/g.

Thank you very much for your suggestions. Now we have revised it in the revised manuscript.

- Line 198: add "of Mohammed"

Thank you very much for your suggestions. Now we have revised it in the revised manuscript.

- Lines 206-207: For greater clarity to the reader, it is recommended to rewrite this sentence.

Thank you very much for your suggestions. Now we have revised it in the revised manuscript.

- Line 209: add "A total"

Thank you very much for your suggestions. Now we have revised it in the revised manuscript.

- Line 229-230: it is recommended to change "In this study, MPPs exhibited considerable hydroxyl radical, superoxide radical, DPPH• radical scavenging activities" to "In this study, MPPs exhibited considerable hydroxyl, superoxide, and DPPH• radical scavenging activities"

Thank you very much for your suggestions. Now we have added this information in the revised manuscript.

- Line 289: Add "s" to "mechanismS"

Thank you very much for your suggestions. Now we have revised it in the revised manuscript.

- Line 302 Add "In the case"

Thank you very much for your suggestions. Now we have revised it in the revised manuscript.

- Line 307: add "a previous"; "the interaction"

Thank you very much for your suggestions. Now we have revised it in the revised manuscript.

- Line 309: add "the quenching"

Thank you very much for your suggestions. Now we have revised it in the revised manuscript.

- Line 312: add "the exponential"

Thank you very much for your suggestions. Now we have revised it in the revised manuscript.

- Line 320: choose between "another study" or "other studies"

Thank you very much for your suggestions. Now we have added this information in the revised manuscript.

- Line 323: add "in the microenvironment"

Thank you very much for your suggestions. Now we have revised it in the revised manuscript.

- Line 332: add "are shown"

Thank you very much for your suggestions. Now we have revised it in the revised manuscript.

- Line 351: add "the considerable"

Thank you very much for your suggestions. Now we have revised it in the revised manuscript.

- Line 362: delete "those of"

Thank you very much for your suggestions. Now we have revised it in the revised manuscript.

- Lines 377-381: it is recommended to improve the sentence

Thank you very much for your suggestions. Now we have revised it in the revised manuscript.

- Line 406: change "In this study, the results" to "This study"

Thank you very much for your suggestions. Now we have revised it in the revised manuscript.

This manuscript is a resubmission of an earlier submission. The following is a list of the peer review reports and author responses from that submission.

Round 1

Reviewer 1 Report

This is a very nice work:

  • Methods, section 2.3, please include methods & conditions of UV detector. If PDA was used, it would be also better to include absorption peaks in table 1.
  • Section 2.6 need a citation to the method if any
  • Results: figure 1, it is confusing, that significant starts from b or c, but not a in all figures.
  • Figures should be large enough for readers.

Reviewer 2 Report

The present work is devoted to improving the method for extracting polyphenols from mangosteen pericarp and evaluating their antioxidant activity and ability to inhibit alpha-amylase. The article does not carry significantly new information; moreover, the methodological part raises serious flaws. 

When testing the ability of polyphenols to inhibit amylase, the authors do not use a positive control. In addition, it is not indicated which amylase was used. It is not possible to study the kinetics of an extract rather than a homogeneous substance. 

Molecular modeling is described very scarsely and does not contain information on model validation. 

In the in vivo experiment, there is also no positive control (acarbose for example), and the method of introducing polyphenols mixed with starch does not allow reaching the exact concentration of the test substance. The authors did not conduct a dose-response study.

See more notes in the MS attached.

Reviewer 3 Report

The paper on Inhibitory Effect of Polyphenol Extracts from Mangosteen (Garcinia mangostana) Pericarp on α-Amylase is well written. It reads well and the research has been conceptualized coherently. I only have two suggestions to make:- 

  1. Distinguish the experiments that were in vitro and those that were in vivo on the abstract
  2. Line 2 of the introduction should be rephrased. It reads: In 2019, about 463 million adults worldwide with diabetes (1 of 11 people were diabetic) and about 4.2 million deaths were attributed to diabetes or the relative complications. But it should change to:- In 2019, about 463 million adults worldwide lived with diabetes (1 of 11 people were diabetic) and about 4.2 million deaths were attributed to diabetes or the relative complications

Reviewer 4 Report

This manuscript deals with the extraction of phenolic compounds from Mangosteen Pericarp and mesurement of inhibition activity against α-Amylase. The major drawbacks of this study is that the differences in extraction yields with different extraction conditions are very small, so the experimental errors are too large(Table S2 and Fig. 1(C, D)). Therefore, the new experimental design is recommended.

-What is the difference between extraction yield(Table S2) and extraction ratio(Fig 1).

-The extraction yields (2.65-3.09%) are too small.

-How did you get the extraction yield data? Is it based on the content of PP in the raw sample??

-How much is the PP content in the original sample and how did the author extract it??

-Table S2: the differences in extraction yields with different extraction conditions are very small, so the experimental design was not well done.

-Fig. 1(C, D): the experimental errors are large.